# Green Bees: Reverse Genetic Analysis of Deformed Wing Virus Transmission, Replication, and Tropism

**DOI:** 10.3390/v12050532

**Published:** 2020-05-12

**Authors:** Olesya N. Gusachenko, Luke Woodford, Katharin Balbirnie-Cumming, Ewan M. Campbell, Craig R. Christie, Alan S. Bowman, David J. Evans

**Affiliations:** 1Biomedical Sciences Research Complex, University of St. Andrews, St. Andrews KY16 9ST, UK; lw86@st-andrews.ac.uk (L.W.); d.j.evans@st-andrews.ac.uk (D.J.E.); 2Centre for Inflammation Research, Queen‘s Medical Research Institute, University of Edinburgh, Edinburgh EH16 4TJ, UK; kebalbirnie@outlook.com; 3Institute of Biological and Environmental Sciences, School of Biological Sciences, University of Aberdeen, Aberdeen AB24 3FX, UK; e.m.campbell@abdn.ac.uk (E.M.C.); craig.christie@abdn.ac.uk (C.R.C.); a.bowman@abdn.ac.uk (A.S.B.)

**Keywords:** insect viruses, honey bee, pollination, virus vector, *Varroa*, RNA viruses, DWV, reverse genetics

## Abstract

Environmental and agricultural pollination services by honey bees, *Apis mellifera*, and honey production are compromised by high levels of annual colony losses globally. The majority are associated with disease caused by deformed wing virus (DWV), a positive-strand RNA virus, exacerbated by the ectoparasitic mite *Varroa destructor*. To improve honey bee health, a better understanding of virus transmission and pathogenesis is needed which requires the development of tools to study virus replication, transmission, and localisation. We report the use of reverse genetic (RG) systems for the predominant genetically distinct variants of DWV to address these questions. All RG-recovered viruses replicate within 24 h post-inoculation of pupae and could recapitulate the characteristic symptoms of DWV disease upon eclosion. Larvae were significantly less susceptible but could be infected orally and subsequently developed disease. Using genetically tagged RG DWV and an *in vitro Varroa* feeding system, we demonstrate virus replication in the mite by accumulation of tagged negative-strand viral replication intermediates. We additionally apply a modified DWV genome expressing a fluorescent reporter protein for direct *in vivo* observation of virus distribution in injected pupae or fed larvae. Using this, we demonstrate extensive sites of virus replication in a range of pupal tissues and organs and in the nascent wing buds in larvae fed high levels of virus, indicative of a direct association between virus replication and pathogenesis. These studies provide insights into virus replication kinetics, tropism, transmission, and pathogenesis, and produce new tools to help develop the understanding needed to control DWV-mediated colony losses.

## 1. Introduction

Deformed Wing Virus (DWV) is arguably—in concert with its vector the ectoparasitic mite *Varroa destructor*—the most important pathogen of the European honey bee (*Apis mellifera*). DWV has a near-global distribution (excluding Australia, where it is either absent or present at lower levels without *Varroa* transmission [1]) and, in the absence of *Varroa*, persists at low levels and is rarely pathogenic [2]. In contrast, when transmitted by mites, DWV titres become highly elevated, infested pupae may develop characteristic symptoms and significant levels of overwintering colony losses occur [3,4,5,6], attributable to the reduction of honey bee longevity [7]. Evidence on the selective evolution of DWV through vector transmission bottlenecking [8,9], selection of *Varroa* propagating DWV variants [10], and synergistic action of the mite and the virus on the host [11] have been reported. However, there remain conflicting studies on the ability of the mite to support DWV replication with some indicating biological [10] and others favouring non-propagative transmission routes [12]. Although there is a clear correlation between the mite-borne transmission and symptomatic outcome of the DWV infection the underlying mechanisms of the cooperative action of the two pathogens needs further clarification. A better understanding of DWV pathogenesis is needed to further develop intervention strategies to prevent and control disease. 

DWV is a picorna-like virus from the *Iflaviridae* family [13,14]. The single-stranded positive-sense RNA genome encodes a polyprotein flanked by 5′- and 3′-untranslated regions (UTR). Based upon our understanding of related viruses, the polyprotein is processed by viral and/or cellular enzymes into the structural and non-structural proteins required to complete the virus life cycle. The structural proteins form the virus capsid [14], whereas the non-structural proteins modify the cellular milieu and replicate the genome. Like other RNA viruses, DWV is genetically diverse, with a related complex of viruses divided into two or three groups sharing ~84–97% genetic identity. DWV A [14] and Kakugo virus [13] exhibit 97% identity in their RNA sequences and form the type A subgroup. Another master variant of DWV was initially isolated from *Varroa* and named Varroa Destructor Virus type 1 (VDV-1) [15]. As a consequence of its high sequence similarity (84/95% identity at the RNA and protein levels respectively to DWV A) [16] and its ability to infect the same host (honey bee), it is often referred to as DWV type B [17,18]. A third master variant of the virus designated as DWV C has also been reported [18]. A range of differences in host preference, tissue tropism, morbidity, and pathogenicity have been suggested for the two master variants [9,10,19,20,21,22]. For example, the predominance of DWV A in a landscape-scale study on Hawaii following the introduction of *Varroa* to naïve colonies with a diverse virus population was interpreted as an indication that this variant was more virulent [17,22]. Conversely, in side-by-side studies in laboratory experiments, DWV A had a less pronounced effect on adult honey bee survival compared to DWV B or a mixture of both variants [20]. Further studies using field sourced inoculates of DWV A and B showed that they were equally virulent and generated similar levels of morbidity in emerged adult bees [23]. In addition to these so-called master variants, a range of recombinants between DWV A and B have been reported [21,24,25,26]. For example, VDV-1_DVD_ (GenBank HM067437) and VDV-1_VVD_ (VDV-1-DWV-No-9, GenBank HM067438), both bearing the DWV A capsid proteins coding region and DWV B non-structural coding region [26]. In some studies, these accumulated to a higher level in infected honey bees than the parental strains and it has been suggested that evolution of the DWV quasispecies is driven by *Varroa* transmission toward the emergence of variants with enhanced virulence [21,26]. All of these reports are based on virus field isolates, and it remains unclear whether the DWV master variants and recombinants fundamentally differ in their phenotypes or if the differences reported reflect local strain variation or the experimental system used [9,17]. Therefore, further studies are required to associate the virulence with a particular genotype. A direct way to address this, and one that allows the propagation of near-clonal viral stocks for analysis, is to generate viruses using a reverse genetic (RG) system. 

In virology, RG involves the manipulation of the genotype, the recovery of the virus, and the investigation of the phenotype. Over almost four decades, it has become the *de facto* standard approach to address questions about virus replication, virulence and pathogenesis [27,28]. To facilitate these studies, a range of genome modifications (e.g. reporter genes) have been used to allow the sensitive quantification and localization of the virus [29]. Molecular cloning of individual genetic variants of DWV is required to establish a direct connection between infection, and the observed symptoms. RG systems for type A DWV have been reported [12,30]. In this study, we exploit an extended RG toolbox for both DWV A and B master variants and a type B/A recombinant to investigate their comparative transmission, tropism and pathogenesis in honey bees. Using these resources, we also provide direct evidence of DWV replication in *Varroa destructor*. Finally, by introducing a reporter encoding sequence to the virus genome we visualise the *in vivo* tissue distribution of DWV in infected honey bee brood. The results of this study provide new insights into understanding the nature of DWV infection and introduce new molecular tools for honey bee research.

## 2. Materials and Methods 

### 2.1. RG System for Three DWV Variants

RG constructs used in this study were based on the cDNA clone of a recombinant DWV variant VDV-1-DWV-No-9 (GenBank HM067438.1). VDV-1-DWV-No-9 sequence was rescued using samples from a *Varroa*-infested honey bee colony from Warwick-HRI apiary (2014), and a cloned full-length cDNA was incorporated into a plasmid vector containing all required elements for the transcription of the viral RNA. VVD RG clone is identical to the source VDV-1-DWV-No-9, with the exception of two nucleotide substitutions (positions 5277 and 5280 in the VVD clone cDNA, which corresponds to positions 5124 and 5127 in the GenBank HM067438.1 sequence lacking the very 5‘-end of the virus genome), resulting in creation of an *HpaI* restriction site. In order to obtain VDD and VVV constructs bases 2727–4888 (capsid proteins encoding region) and 4885-9783 (non-structural proteins encoding region) were replaced with corresponding DWV A and B fragments respectively (Figure 1, Appendix A). Inserts encoding DWV A capsid proteins and DWV B non-structural proteins were based on published data [14,15] and obtained by custom gene synthesis (IDT, Leuven, Belgium). The replacement sequences were amplified with High Fidelity Phusion DNA polymerase (Thermo Fisher Scientific, Winsford, UK) and incorporated into the initial construct using NEBuilder Hifi assembly reaction (New England Biolabs, Hitchin, UK). The nomenclature used for the corresponding virus clones in this study is as follows: “VDD” —type A DWV (protein coding part), “VVV”—type B, “VVD”—B/A recombinant (see Figure 1 for details). New restriction sites were introduced into each plasmid either by standard site-directed mutagenesis or by including the modification into the synthetic sequence: *HpaI* (5275-5280 nt in VVD variant), *Kpn2*I and *AvrII* (2751-2756 and 4884-4889 nt in VDD variant), and *AvrII*, *PflFI* and *BglII* (4884-4889, 6087-6095, and 9783-9788 nt, respectively, in VVV variant).

EGFP-encoding RG constructs (DWV_E_) were obtained by NEBuilder Hifi assembly (New England Biolabs, Hitchin, UK) reaction applied to previously made RG constructs. EGFP sequence flanked by 32 additional nucleotides (ATGGATAACCCT from 5’ and GCAAAACCAGAG from 3’ end) resulting in duplication of the protease cleavage site (amino acid composition of the site is AKPEMDNP [14]) was inserted between nucleotides 1785–1786 of DWV cDNA. All plasmids were subjected to Sanger sequencing and the results were aligned with data from *in silico* cloning simulation. Full cDNA sequences of DWV clones used in this study are available in Appendix A (Appendix A) and online (GenBank accession numbers: DWV-VDD - MT415949, DWV-VVD - MT415950, DWV-VVD_truncated - MT415951, DWV-VVV - MT415952, DWV-VDD-eGFP - MT415948, DWV-VVD-eGFP - MT415953).

### 2.2. Viral RNA Synthesis

DWV RNA was synthesized using linearized plasmid templates. Full length and truncated templates were linearised with *Pme* I (cutting at the end of the sequence encoding the poly-A tail) or *Nru* I (nt 9231 located within the sequence encoding the viral polymerase) respectively. Linearized DNA was purified by phenol-chloroform extraction and ethanol precipitation and T7 transcription was performed with T7 RiboMAX^TM^ Express Large Scale RNA Production System (Promega, Southampton, UK) according to the manufacturer’s protocol. In order to account for the stabilization effect of the poly-A tail truncated transcripts were subjected to an additional polyadenylation step with poly(A) Tailing Kit (Thermo Fisher Scientific, Winsford, UK). RNA transcripts were purified with GeneJet RNA Purification Kit (Thermo Fisher Scientific, Winsford, UK) using clean up protocol and eluted in RNAse free H_2_O. All transcripts were analysed for integrity by gel electrophoresis and stored at −80 °C.

### 2.3. Virus Stocks

DWV stocks were prepared from honey bee pupae injected with *in vitro* transcribed RNA. Homogenized tissue was diluted with sterile PBS in 1:1 (*w*:*v*) ratio and centrifuged at 13 000× *g*, 4 °C for 10 min. The supernatant was sterilized by passing through 0.22 μM PES filter (Merck Millipore, Watford, UK) and treated with RNase A to destroy all non-encapsidated RNA. RNA was extracted from 100 μL of the virus stock using RNeasy kit (Qiagen, Manchester, UK) and analysed by RT-qPCR.

### 2.4. Honey Bees

All honey bee (*Apis mellifera*) brood in this study was obtained from the research apiary of the University of St Andrews. Hives used for sampling were regularly treated for *Varroa* with an appropriate miticide and routinely screened for DWV. DWV level in bees obtained from these hives was found to range within 10^2^–10^6^ genome equivalents (GE) per 1 μg of total RNA. No phenotypic evidence for other honey bee viruses being present was found within the colonies throughout the course of the studies. Pupae and larvae of the required age were collected from the comb and transferred to the incubator set at 34.5 °C and 90% relative humidity. Pupae were kept on folded sheets of filter paper, and larvae were transferred into 96-well plates with round bottom wells containing feeding diet (6% (*w*/*v*) glucose, 6% (*w*/*v*) fructose, 1% (*w*/*v*) yeast extract, and 50% (*w*/*v*) royal jelly in H_2_O).

### 2.5. Varroa Feeding

*Varroa* mites were collected from the brood frames of infested colonies kept at the University of Aberdeen. Mites were placed in Petri dishes in groups of 10 with a single honey bee pupa and kept overnight in the incubator (34.5 °C, 90% humidity). On the next day, feeding packets containing artificial feeding diet and supplemented with DWV stock or equal amount of PBS were prepared. The feeding packets were prepared by wrapping a 175 μL drop of the diet containing RNase A treated DWV inoculate (prepared as described in Section 2.3). The diet is 75:25 holidic:locust haemolymph (the detailed description of the feeding diet to be published separately [31]). To prepare the diet the locust (fourth instar *Locusta migratoria*) haemolymph was frozen-thawed, heat-treated, and centrifuged for 30 s at 7500× *g*.

Mites were placed in groups of 5–10 mites per feeding packet with four replicates for each treatment and kept in the incubator for four days. At the end of incubation live mites were collected, snap-frozen in liquid nitrogen and stored at −80 °C until further analysis. Total RNA was extracted from pooled mite samples using the standard TriReagent protocol (Thermo Fisher Scientific, Winsford, UK).

### 2.6. RNA Injections and Virus Inoculations

RNA and virus injections into pupae were performed with insulin syringes (BD Micro Fine Plus, 1 mL, 30 G (Becton Dickinson, Oxford, UK)). 5 μL injections were introduced by inserting the syringe needle between the second and third abdominal segments of a pupa. All pupae were maintained on a warm heating plate during the injections.

0.5 and 5 μg (equivalent to 8.7 × 10^10^ and 8.7 × 10^11^ of DWV RNA copies) of *in vitro* transcribed RNA was injected individually into white-eyed honey bee pupae. Truncated VVD transcript injections were used as a negative control in RNA transcript injections. RNA-injected pupae were analysed at 72 h post-injection. 

Injections of pupae with virus stocks were performed using the same technique as for RNA transcripts. Serial dilutions of DWV stocks in sterile PBS were prepared immediately before the injections. Mock control groups were injected with PBS only, while non-injected controls were left intact throughout the duration of the experiment.

For oral infection, larvae were placed into 96 well plates with a diet supplemented with DWV. Fresh diet without virus was added after 24 h or when all virus-supplemented diet was consumed.

### 2.7. RNA Extraction, Reverse Transcription and PCR

Total RNA was extracted individually from all bee samples, *Varroa* mites were analysed in pools of 5–10 according to the treatment group. Samples were homogenized using a Precellys Evolution instrument (Bertin Instruments, Montigny-le-Bretonneux, France). Total RNA was extracted with GeneJet RNA Purification Kit (Thermo Fisher Scientific, Winsford, UK) using a protocol adapted for vacuum manifold application. cDNA was prepared with qScript cDNA Synthesis Kit (Quanta Biosciences, VWR International Ltd, Lutterworth, UK) from 1 μg of total RNA following the manufacturer’s protocol with both oligo dT and random hexamer primers included in the reaction mixture in the reaction volume of 20 μL.

All sequences of primers used in this study are shown in Appendix A. Detection of DWV and honey bee *actin*, used as an internal RNA quality control, was carried out by end-point PCR with *Taq* DNA polymerase (New England Biolabs, Hitchin, UK) and 2 μL of cDNA. DWV_RTPCR primers were designed to detect all three DWV variants under study. To amplify the cDNA regions containing restriction site tags in VDD and VVV virus variants Kpn2I_F/R and PflFI_F/R primer sets were used respectively. PCR cycling conditions were 30 cycles of 95 °C (15 s), 55 °C (15 s), 68 °C (2 min) with an initial 95 °C step (30 s), and a final extension at 68 °C (5 min). PCR samples were analysed on a 1% agarose gel stained with ethidium bromide. When required, DWV PCR products were subjected to restriction digest prior to loading on the gel.

The quantification of DWV genome copies was performed by SYBR-Green Real-Time Quantitative PCR (qPCR). Reactions were carried out in a C1000 Thermal Cycler (Bio-Rad Laboratories, Deeside, UK) using Luna Universal qPCR master mix (New England Biolabs, Hitchin, UK), 0.25 μM forward and reverse DWV_qPCR primers, and 2 μL of cDNA with the following thermal profile: 1 min at 95 °C, followed by 40 cycles of 15 s at 95 °C and 30 s at 60 °C with a final post-amplification melting curve analysis step. Accumulation of DWV_E_ RNA was assayed by a set of primers amplifying a junction region between EGFP and viral sequence (primers EGFP_qPCR_F and VDD_VP2qPCR_RP or VVV_VP2qPCR_RP depending on the virus variant used). DWV and DWV_E_ titers were calculated by relating the resulting Ct value to the standard curve generated by performing qPCR from a serial dilution of the cDNA obtained from 1 μg of VVD or DWV_E_ RNA transcript respectively.

### 2.8. Negative Strand Assay

Strand-specific detection of DWV RNA was performed as described earlier [10]. Briefly, 1 µg of total RNA was used in reverse transcription reaction carried out with Superscript III reverse transcriptase (Invitrogen, Thermo Fisher Scientific, Winsford, UK) and the adapter extended primer DWV(-RNA)_RT designed to anneal to the negative strand RNA of DWV. The PCR step was carried out by *Taq* DNA polymerase (New England Biolabs, Hitchin, UK) using a forward primer identical to the adapter sequence (primer 388) and DWV(-RNA)_RT R primer. PCR was run for 35 cycles in the same conditions as described above.

### 2.9. Microscopy

All imaging to detect EGFP signal in samples infected with DWV_E_ was done using Leica TCS SP8 confocal microscope with 10× HC PL FLUOTAR objective (Leica Biosystems, Newcastle, UK). 

### 2.10. Cryosection of Larvae Samples

Live larvae were washed with increasing concentrations of aqueous ethanol solution and fixed in 4% paraformaldehyde in PBS for 2 h at 4 °C. Fixed larvae were allowed to sink in 30% sucrose in PBS, mounted in NEG-50 Frozen Section Medium (Thermo Fisher Scientific, Winsford, UK) and subjected to microsectioning on CM1860 Cryostat (Leica Biosystems, Newcastle, UK). Sections of 50–80 µm thickness were placed on Superfrost Plus microscope slides (Thermo Fisher Scientific, Winsford, UK), mounted in ProLong Gold antifade medium, and analysed by microscopy.

## 3. Results

### 3.1. Infectivity of Parental and Recombinant DWV Variants in Honey bee Brood

To examine the biology and pathogenesis of different strains of DWV we prepared near-clonal virus stocks from honey bee pupae injected with *in vitro* transcribed viral RNA from relevant cDNAs. All RG templates were derived from an infectious cDNA of a recombinant DWV variant (GenBank HM067438.1) [26]. The 5′ UTR, capsid-coding, and 5′ part of the presumed helicase-coding region of this genome are 99% identical to VDV-1 (DWV B, AY251269.2) with the remainder of the genome ~97% identical to DWV-A (NC_004830.2) henceforth this variant was designated VVD. The VVD cDNA was modified by replacing the capsid-coding region with the analogous sequence from DWV A to generate a VDD cDNA, and by replacement of the 3′ non-structural protein coding region with DWV B sequence to generate VVV cDNA (Figure 1) [32]. Each cDNA carried unique synonymous genetic tags (restriction sites) allowing their unambiguous identification. The protein coding part of the resulting VDD and VVV constructs was 98.51% and 99.44% identical to the reference DWV A and DWV B field genomes respectively (99.65% and 99.79% identity at the encoded protein level).

Direct inoculation of white-eyed honey bee pupae with *in vitro* synthesised RNA enabled recovery of genetically tagged virus at high efficiency. Using qPCR analysis, the average viral load in RNA-injected pupae was shown to be 1.5 × 10^10^ DWV GE per 1 μg of total RNA (3 × 10^12^ GE per pupa, Appendix A) compared to ≤10^5^ DWV GE/μg of RNA in mock-inoculated pupae. Therefore at least 99.999% of the virus preparations from injected pupae originated from the cDNA-derived injected RNA. In each case, the identity of the virus was verified by RT-PCR and the presence of the unique restriction endonuclease site engineered into the cDNA was confirmed.

We investigated the infectivity of RG-recovered VDD, VVV and VVD variants by inoculation of white-eyed honey bee pupae using serial dilutions of virus stocks and analysed all samples individually 24 h post-inoculation by RT-qPCR (Figure 2a and Appendix A). Under these conditions, we observed robust accumulation of virus following inoculation of just 1 GE, with increasing yields of virus when 10–1000 GE were injected. At the lowest level of inoculum VDD appeared to accumulate slightly more slowly (Tukey’s multiple comparisons test, *p* = 0.0319 for VVD vs VVD in 1 GE injections groups), but there was no discernible difference in yield of the three variants when ≥10 GE were inoculated (*p* > 0.05). Incubation of pupae for a further 24 h resulted in virus yields of ~10^10^ GE/μg of RNA with no notable difference between the three DWV variants tested. This shows that RG-generated DWV is infectious and the kinetics of virus replication is very rapid, with the genome being amplified ~10^8^–10^13^ times within 48 h of injection of pupae (minimum yield of RNA/pupa was 200 μg).

*Varroa* feed throughout the development of the pupa and possibly during the phoretic phase of the life cycle. We investigated whether pupal development phases were partially or totally refractory to DWV replication by inoculating morphologically distinct stages with 10^2^ GE of VVD DWV (Figure 2b). In each instance DWV levels reached ~10^9^ GE/μg within 24 h post-injection, indicating that all stages of pupal development are apparently equally capable of supporting DWV replication.

Having demonstrated pupal susceptibility to directly inoculated virus we then examined virus transmission to developing larvae via the oral route. First instar larvae were individually fed a diet containing dilutions of RG-derived DWV, and the virus was detected and quantified after 48 h. Although RG-derived DWV was detectable when 5 × 10^5^ GE were fed, virus replication to levels distinctly higher than the inoculum required an input of at least 5 × 10^6^ GE, with VDD again slightly slower than the other variants tested (Figure 2c). Uninoculated control larvae contained ~10^5^ DWV GE/μg of RNA, presumably reflecting virus acquired during initial feeding in the hive or vertically from the queen. All larvae in DWV-fed groups contained elevated levels of viral RNA compared to the control group 48 h after inoculation (Tukey’s multiple comparisons test, *p* = 0.0372 for VVD vs mock, *p* = 0.0205 for VDD vs mock, for 5 × 10^5^ GE feeding groups). High DWV levels (up to 10^10^ GE/μg of RNA) were detected in 5 × 10^6^ GE fed groups for VVV and VVD inoculates and for all three variants in 5 × 10^7^ GE fed larvae, clearly indicating virus replication. It was notable that a proportion of larvae showed little or no amplification of DWV over the fed amount suggesting, in comparison to inoculation via injection, a significant barrier to the development of a productive infection may exist for virus acquired *per os*. 

### 3.2. Replication of RG DWV in Varroa Destructor

There are conflicting reports on the replication of DWV in *Varroa*. A confounding issue in some studies is the pre-existing presence of DWV within the mite, and the potential presence of contaminating DWV genomic negative strands (a marker of replication) in either the mite or the inoculum. To address this, we investigated the replication of RG-derived genetically tagged VVD in *Varroa* using an *in vitro* feeding system [31] containing no honey bee-derived material (Figure 3a).

VVD RG-derived virus stock was amplified in honey bee pupae, extracted and treated with ribonuclease A to remove any non-encapsidated viral or cellular RNA. *Varroa* were maintained on artificial feed packets supplemented with 1.75 × 10^9^ GE VVD (10^7^ GE/μl). 

After four days, mites were harvested, total RNA was extracted and screened using a DWV negative strand-specific RT-PCR assay. All pooled mite samples from DWV-fed groups produced a DWV-specific product; three out of four mite pools from the mock-inoculated groups also produced a product as expected since these were harvested from hives with high *Varroa*, and consequently high DWV, levels. Upon digestion with the relevant restriction enzyme (*HpaI*), this product was partially cleaved in the VVD groups only, indicating that at least some of it originated from the negative-strand replication intermediate of VVD (Figure 3b). Importantly, no negative strand RNA of DWV was detected in the RNAse-treated input virus stock used for these feeding experiments. The absence of negative RNA strands in the viral input combined with the specific detection of RG-tagged DWV negative strand RNA in mites that have fed provides evidence for the replication of VVD DWV in *Varroa destructor*.

### 3.3. Pathogenicity of DWV Variants in Honey Bees

DWV infection in honey bees is known to result in a range of developmental abnormalities, the most prominent of which is malformation or arrested unfolding of wings [14,33,34,35]. Since RG-recovered DWV replicated to high levels in injected pupae, we went on to determine whether pupae incubated until eclosion also exhibited characteristic developmental defects. DWV inoculation at the white-eyed pupal stage, mimicking the route and timing of *Varroa* transmission [9,30,36], resulted in 75–80% of bees developing overtly deformed wings (Figure 4b). Analysis by qPCR showed no difference in the final DWV titers between highly deformed and apparently phenotypically normal eclosed workers (Figure 4a). In the same experiment we observed no differences in either virus levels or the proportions of normal vs deformed workers with the three different RG DWV variants tested. These findings indicate that RG-derived DWV is pathogenic when directly inoculated into developing worker pupae and results in symptoms and virus titres that are similar to those seen in *Varroa*-exposed pupae [9]. In addition, this study demonstrated that phenotypically normal eclosed workers can have virus levels indistinguishable from those with deformed wings. This confirmes that wing deformities are not an inevitable consequence of high levels of DWV replication [23,30].

In the field non-*Varroa* transmission of DWV (*per os* or vertically) results in predominantly asymptomatic (also referred to as covert) infection, with no apparent phenotypic deformities. We tested the morbidity in honey bees infected orally with clonal DWV by feeding larvae with a virus-supplemented diet, allowing pupation and eclosion and scoring viability and the presence of overt symptoms. Although lower levels of virus (≤10^5^ GE) in the diet again failed to establish robust replication in the experimental groups, the feeding of 2 × 10^7^ GE of RG-derived DWV to honey bee larvae (delivered as single time feeding at the first instar larval stage) resulted in high morbidity rates (Appendix A). In larval groups infected with DWV within 24 h of hatching from eggs, all eclosed adult honey bees had deformities ranging from malformation of one or both wings, abdominal bloating, discoloration, and dwarfism. A proportion also exhibited arrested development at the pupal stage and failed to eclose. The majority of the developed bees revealed high virus levels of up to 10^12^ GE/μg of RNA (Appendix A). Notably, the laboratory-reared honey bee brood revealed relatively high levels of wing deformities present in mock-inoculated pupae and PBS-fed larvae upon eclosion (Figure 4b and Appendix A). It is known that alterations in incubation conditions of pupae at key stages of development can also result in wing deformities [30], suggesting that wing morphogenesis is a particularly sensitive stage of development.

### 3.4. DWV Localization in Infected Honey Bee Brood

The tropism and pathogenesis of DWV remains poorly understood. Relatively little is known about the sites of virus replication and whether differences in tissue tropism after oral or mite-mediated transmission account for the appearance of symptomatic and asymptomatic forms of disease. To facilitate virus localization in experimentally inoculated bees, we developed an EGFP-expressing chimeric DWV (designated DWV_E_), analogous to that previously reported for DWV A [37]. Briefly, the cDNA encoding the viral polypeptide was modified by the insertion of the EGFP cDNA at the junction between the Leader and VP2 capsid protein coding regions similarly to the system developed earlier for the poliovirus [38]. The in-frame EGFP sequence was flanked with partial duplications of the predicted 3C^pro^ proteolytic cleavage site [14] with the intention of co-translational processing and release of the EGFP protein (Figure 5a).

Injection of *in vitro*–synthesised DWV_E_ RNA into white-eyed honey bee pupae resulted in distinct green fluorescence in the head, thorax and abdomen observable within 20 h post inoculation (Figure 5b). The fluorescence signal could be readily detected throughout inoculated pupae at least seven days post-injection (Appendix A), implying that this represents a robust experimental system to investigate virus localization *in vivo*. In preliminary studies we went on to investigate the tissue tropism of reporter-encoding derivates of VDD, VVV and VVD variants of DWV in inoculated pupae and observed no discernible differences; therefore, subsequent analysis was performed using the EGFP derivative of VDD.

Confocal microscopy allowed the visualisation of individual foci of virus replication, identified as distinct punctate fluorescence, in a range of tissues throughout the inoculated developing pupae. At 22 h post inoculation the fluorescent signal was apparent in the head, parts of the gut (the crop, ventriculus, small gut and rectum), but was largely absent from the Malphigian tubules and thoracic muscles (Figure 5c). Since overt DWV is predominantly associated with wing deformities we looked in detail at the developing wings in inoculated pupae. At 22 h post-inoculation punctate fluorescence was clearly visible in the wings. White-eyed pupae inoculated with DWV_E_ and incubated until eclosion were also examined. As before (Figure 4), all had significant levels of virus replication irrespective of the presence of overtly deformed wings. The wings of 90% of all injected pupae and honey bees developed from these pupae, regardless of their deformity status, were shown to express EGFP upon eclosion indicative of sites of DWV_E_ replication (Appendix A).

To trace DWV distribution following oral acquisition of virus we fed developing larvae with 5 × 10^7^ GE of DWV_E_ purified from previously inoculated pupae and analysed larval fluorescence 4 and 6 days later. Samples infected with DWV_E_ exhibited extensive fluorescence; external observation localized this to the larval head, two thoracic segments containing wing rudiments, the caudal part of the body and to cells surrounding the spiracle openings (Figure 5d, Appendix A). Additional sites of EGFP localization were apparent after cryosectioning samples, confirming DWV_E_ infection in the larval head, tissues of thoracal and caudal abdominal segments (Appendix A), and in the developing reproductive system (Figure 5d).

## 4. Discussion

The inexorable rise in global honey bee colony numbers masks increasing levels of colony losses, which are regularly reported to exceed 30% per annum and predominantly occur during the winter in temperate regions [39,40,41]. For almost a decade the major cause of these losses, the ectoparasitic mite *Varroa destructor* and the smorgasbord of viruses it transmits, has been well known [7]. Of these, the most important virus associated with overwintering colony loss is DWV. Although DWV is a single stranded, positive sense RNA virus—a group that includes poliovirus which has been dissected at the molecular level for almost four decades—molecular methods to study its biology have developed slowly. One of the major reasons for this is the absence of a usable *in vitro* cell culture system enabling virus propagation [42]. The recent development of an RG system [30] allowing the recovery of infectious virus from a cDNA for the type A variant of DWV provided the first tractable approach to a better understanding of the biology and pathogenesis of DWV.

We report here the extension of the genetic tools to study the biology of DWV including the type B variant (also designated VDV-1) and a recombinant that has previously been reported to predominate in *Varroa*-infested colonies [9,26]. We additionally demonstrate how the RG approach can be exploited to study host-pathogen and host-vector-pathogen interactions, and—with the development of reporter gene-expressing variants of the virus—to study tissue distribution in developing honey bee larvae, pupae, and eclosed adults.

Using standard molecular cloning techniques combined with *in vitro* gene synthesis, we constructed infectious cDNAs for type A (protein coding sequence), type B and a recombinant variant of DWV, recovered molecularly tagged virus after RNA inoculation and investigated the kinetics of virus replication in honey bee pupae. Replication was rapid, amplifying to ~10^8^ GE/μg of RNA within 24 h, and plateauing at ~10^10^ GE per μg of total cellular RNA within 48 h. There were no major differences between the replication rates of the three DWV variants under study, and all stages of pupal development tested appeared equally susceptible to virus infection (Figure 2). This implies that, although pupae are only naturally exposed to mite-borne virus upon capping, even very low amounts of virus inocula introduced by the mite have the capacity to replicate to very high levels before eclosion. The rapid kinetics of DWV replication also means that all progeny mites produced by a single pupa are likely to almost exclusively carry the virus population representative of that introduced by the foundress mite, as these (and not the endogenous virus population) are what are amplified following direct injection.

Horizontal virus transmission within the colony also occurs during larval feeding by nurse bees. We show that larvae exhibit higher resistance to infection with clonal DWV inoculates *per os* when compared with the susceptibility of pupae to injected virus. Greater than 10^6^-10^7^ GE of virus was needed to reliably infect more than 50% of larvae with any DWV variants tested (Figure 2c). This result indicates that high levels of DWV present in the colony due to amplification by *Varroa* parasitized individuals can boost the development of a symptomatic infection state both through pupal infestation by mites and feeding of larvae by diseased nurse bees. A better understanding of this will require the levels of virus transmitted orally, or present in bee bread fed to developing larvae, to be determined. Interestingly, inoculation with VDD clone containing DWV type A structural proteins sequence required higher virus concentrations to achieve efficient infection in larvae. This result may indicate that the 5’ part of the protein encoding sequence of DWV B genome enables higher infectivity either due to enhanced binding and penetration into the target host cells or via other interactions caused by clone-specific secondary or tertiary structures of viral RNA.

As honey bee pupae appear very sensitive to infection by direct injection (recapitulating transmission by *Varroa*), any virus replication in the mite would likely guarantee the transmitted dose would significantly exceed the ID_50_. The viral load in mites has been reported to be high [43] though this may reflect the level of virus in the preceding host pupa rather than replication *per se*. Previous analysis of virus replication in *Varroa* has produced conflicting results. Mass spectroscopy studies failed to detect viral non-structural proteins implying that DWV may not replicate in the mite [44]. Conversely, although detection of viral negative strand replication intermediates [15,45] in mites may indicate replication, they may also reflect carry-over from the previous pupal feed. To address this, we maintained mites *in vitro* on a diet containing no honey bee-derived components. The only DWV virus (and viral negative strand RNA) present would therefore be derived from the last pupa the mites had fed on, together with any subsequent replication in the mite. We supplemented the diet with the RG-derived VVD variant of DWV carrying a unique genetic tag allowing its unambiguous identification. The presence of genetically tagged negative strand RNA of DWV in the pooled mites samples fed on this diet demonstrates that VVD does replicate in *Varroa* and that newly acquired virus can replicate in the presence of a pre-existing virus population in the mite (Figure 3). The latter point is significant as it suggests that the virus population in the mite reflects its historical diet from successive infested pupae, potentially influenced by any differential virus replication in the mite.

Although mite transmission of DWV is associated with overt symptoms such as wing deformities these are not the inevitable consequence of high viral loads [23,30]. Approximately 25% of inoculated pupae had normal wings on eclosion, similar to another recent report [23], despite having viral loads in excess of 10^10^ GE/μg RNA. While apparently developmentally normal, evidence indicates that workers with high viral loads are impaired in foraging ability, cognitive functions, and die prematurely [7,10,46,47]. In contrast to the pupal injections where a significant proportion eclosed and appeared developmentally normal despite high levels of virus replication, no virus-fed larvae which eclosed exhibited normal wings, although >50% reached the adult stage (Appendix A). These observations suggest that the earlier stages of honey bee development may be more susceptible to DWV-mediated damage. Additionally, this signifies that the occurrence of high DWV levels in bees with normal wings in the field may be due to *Varroa*-mediated transmission, while the outcome of oral infection in larvae depends on the amount of virus ingested and results in either benign asymptomatic infection or in high morbidity due to high levels of DWV replication and consequential developmental damage. Apart from horizontal transmission addressed in this study, DWV can be transmitted vertically from an infected queen [48]. Further studies are required to elucidate the exact impact of this route on the phenotypic outcome of the infection.

It remains unclear what determines whether virus exposure results in overt disease or asymptomatic infection [43] although results presented here suggest that the timing of infection (larvae *vs*. pupae) is probably critical. Analysis of whether infection of all stages of pupal development are as likely to result in wing deformities may be informative in this regard, though *Varroa*-mediated virus transmission will initially occur when the foundress mite feeds on the just-capped pre-pupa. One of the factors which can potentially influence the development of overt disease are the sites of virus replication in the developing honey bee. This may result in direct cytopathicity and tissue damage or indirectly by dysregulation leading to damage at remote locations. To address the tissue tropism of DWV, we constructed a modified virus genome co-translationally expressing the green fluorescent protein. Inoculation of pupae, or ingestion by larvae, resulted in distinct fluorescence in a range of organs and tissues. In fed larvae, the EGFP signal accumulated in thoracic segments where the developing wing buds are located [49], and infection of these presumably accounts for the characteristic symptoms of DWV infection. It was notable that larvae fed with large amounts of DWV invariably developed with malformed wings. The developing ovary of worker larvae also appears to be a site of virus replication. Worker ovaries—other than in laying workers—never produce eggs, so further studies will be required to determine whether the ovary of developing queens also shows evidence of virus replication. This, together with the known presence of DWV in drone semen [50], would presumably explain the vertical transmission of virus. 

Our observations indicate that DWV selectively targets organs most favourable for its oral and vertical transmission. Sites of DWV replication colocalize with exocrine glands, which are responsible for the secretion of larval diet components, and digestive tract tissues, from which it can presumably be transmitted in faeces [51] and, following regurgitation, orally. These results support previous studies using anti-capsid antibodies or riboprobes targeting the virus genome [30,35] where DWV-specific signals were found in honey bee brains, exocrine glands, midgut, fat bodies, and reproductive organs [30,35,52,53]. Since the emergence of *Varroa* as a vector the oral route no longer plays the decisive role in DWV spread, but the observed tropism of DWV infection indicates the initial evolutionary trait used by the virus. Robust virus replication in the primary tissues that enable subsequent transmission often results in spillover to secondary sites that are permissive for virus infection, but that offer no further route to a new host, e.g., the neurotropism of the faecal-orally transmitted poliovirus [54]. It is therefore unsurprising that DWV infection was additionally found in a range of sites in the developing larva or pupa, including the spiracles and nascent wing tissues. The availability of EGFP-markers for sites of virus replication will enable further *in vivo* time course studies of virus dissemination and facilitate host-vector transmission studies. More generally, the ability to introduce a ‘payload’ to the virus genome may also be exploited by using genetically modified DWV as a virus-based gene delivery system [37]. Our results clearly demonstrate that DWV is abundant in many tissues in honey bees and a EGFP, or similar reporter, would enable tissue-specific RNAi responses to be quantified and optimized. Likewise, reporter-expressing derivatives of DWV have potential in tropism and transmission studies in other species in which the virus is known or suspected of replicating, including *Varroa* or *Bombus* [32] or, more speculatively, as gene delivery vectors for mite control.

In many single stranded positive sense RNA viruses, the development of reverse genetic systems was facilitated by a good understanding of virus replication *in vitro*. With no system for propagating DWV *in vitro* these studies have had to be conducted *in vivo*. We show that the availability of a reverse genetic system allows the kinetics of virus replication in larvae and pupae to be examined. Comparative studies demonstrate that the two predominant variants of DWV essentially replicate equivalently, and that the symptomatic outcome of DWV infection is not linked to a particular genotype of the virus nor to the route of transmission. Unique genetic tags introduced to the genome enabled discrimination of input from the endogenous virus population and provide unequivocal evidence for virus replication in *Varroa* following detection of *de novo* synthesised negative-sense viral RNA. A viable GFP-tagged DWV genome supplies the basis for a better understanding of virus tropism and pathogenesis in infected brood and will provide a useful tool for elucidating the mechanisms behind symptomatic DWV infection in developing honey bees.

## Figures and Tables

**Figure 1 viruses-12-00532-f001:**
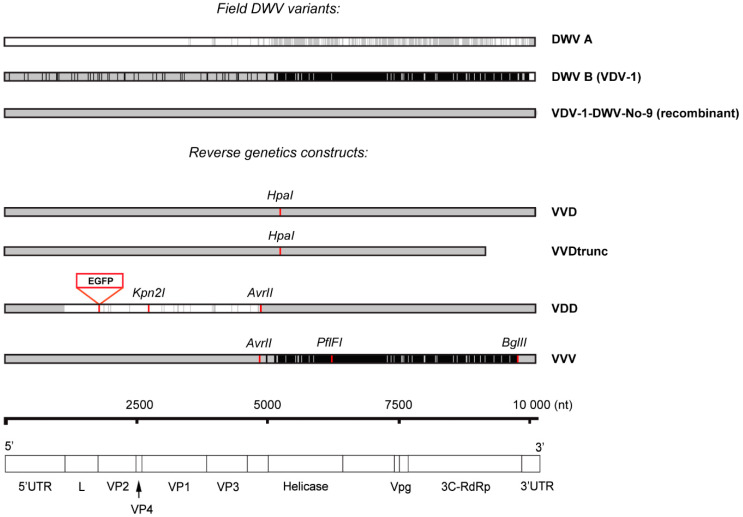
Sequence homology between field DWV type A (NC_004830.2), DWV type B (GenBank: AY251269.2), and recombinant VDV-1-DWV-No-9 (GenBank HM067438.1) variants and RG virus clones. VVD RG clone was prepared using full-length cDNA of VDV-1-DWV-No-9. VDD and VVV RG clones were obtained by replacing 5’ and 3’ parts of protein-encoding sequence of VVD construct with synthetic fragments homologous to corresponding parts of DWV A and DWV B genomes. DWV A and B specific sequences are shown in white and black respectively, genome regions identical to the recombinant clone sequence (VDV-1-DWV-No-9) are shaded in grey. Unique synonymous mutations introducing new restriction sites are indicated in red, as well as the position at the junction between L- and VP2 proteins coding sequence where EGFP insert was incorporated in order to obtain reporter-expressing DWV. Genomic RNA organization of DWV and encoded proteins are shown below with numbers (nt) indicating nucleotide position along the genome.

**Figure 2 viruses-12-00532-f002:**
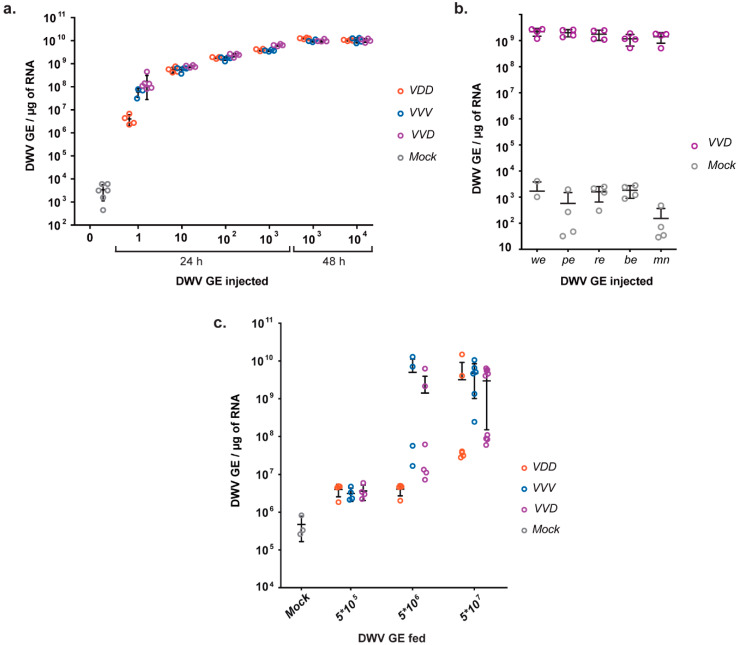
Inoculation of the honey bee brood with RG-DWV. (**a**) RT-qPCR analysis of DWV level in honey bee pupae (injected at white-eyed stage) 24 h and 48 h post-injection with different amounts of DWV variants, “Mock”—uninoculated pupae; (**b**). RT-qPCR analysis of honey bee pupae, injected with 10^2^ GE of VVD at different stages of pupal development: white-eyed (“we”), pink-eyed (“pe”), red-eyed (“re”), blue-eyed (“be”), and at the start of full melanisation (“mn”). Pupae were sacrificed 24 h post-inoculation; (**c**) RT-qPCR analysis of DWV accumulation levels in honey bee larvae 48 h after feeding with DWV variants at different concentrations. Each value corresponds to an individual pupa or larva analysed, and error bars show mean ±SD.

**Figure 3 viruses-12-00532-f003:**
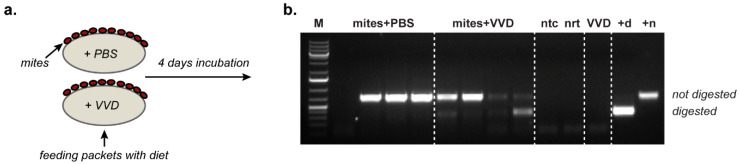
DWV replication in *Varroa* mites. (**a**) *Varroa* feeding experimental setup using feed packets with artificial diet supplemented with RG VVD DWV in PBS or PBS only (control); (**b**) 1% agarose gel with *HpaI*-digested PCR products amplified using a DWV negative strand-specific RT-PCR assay. Specific (-)RNA PCR products from *Varroa* mites fed with artificial diet supplemented with VVD virus stock (“mites+VVD”) or PBS (“mites+PBS”); four groups of 10 mites were used for each feeding condition - each lane corresponds to a pooled sample of mites from one group (fed on the same diet packet), “ntc” and “nrt”—no template and no RT controls, respectively, “VVD”—negative strand-specific RT-PCR analysis of the virus stock used for mite feeding, “ + d” and “+n”—digested and undigested positive PCR controls, “M”—molecular weight DNA marker.

**Figure 4 viruses-12-00532-f004:**
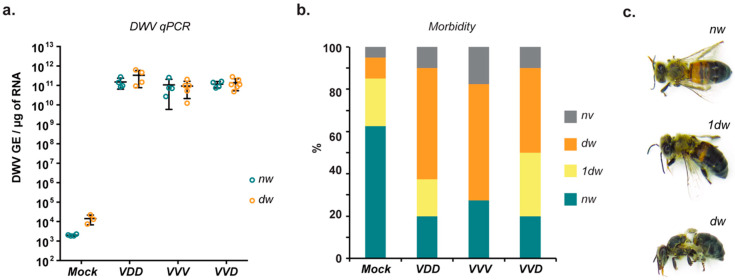
Morbidity of DWV variants in honey bee pupae. (**a**) RT-qPCR analysis of DWV levels in honey bees developed from pupae injected with the indicated virus variants and displaying normal (“nw”) or deformed wing (“dw)” phenotype; no significant difference in DWV levels was found between deformed and non-deformed bees (Sidak’s multiple comparisons test, *p* = 0.1307 (VDD), *p* = 0.9975 (VVV), *p* = 0.9911 (VVD)); (**b**) Percentage of visually normal (“*nw*”), partially deformed (“1dw”), deformed (“*dw*”) and non-viable (“*nv*”) honey bees eclosed from pupae injected at the white-eyed stage with 10^2^ GE of the indicated DWV variants (*n* = 40 for each group); (**c**) Newly emerged honey bees with different phenotypes of morbidity: “nw”—non-deformed bee with fully unfolded wings; “1dw”—bee with only one normal wing; “dw”—bee with deformities in both wings.

**Figure 5 viruses-12-00532-f005:**
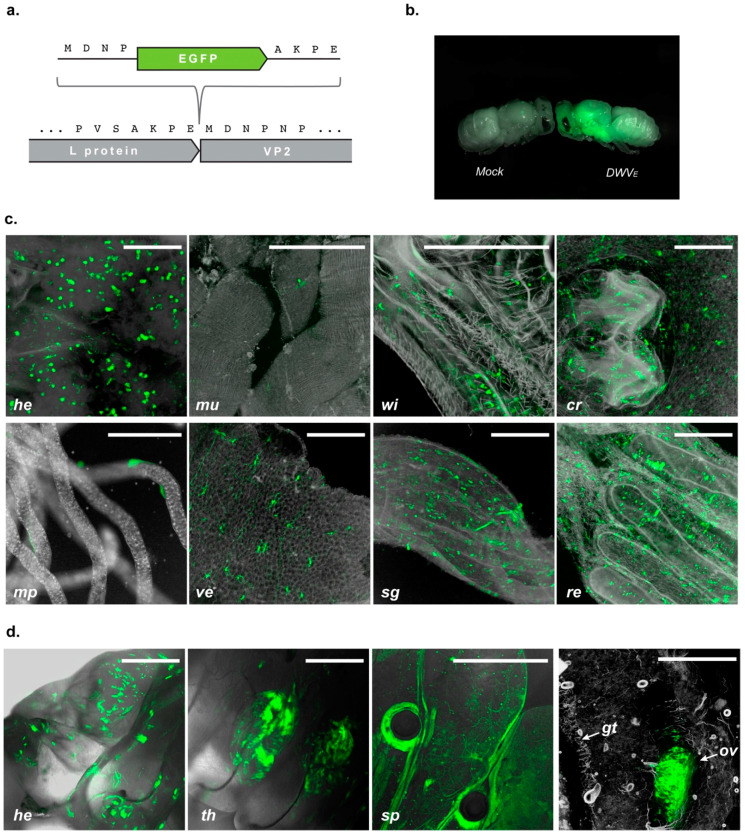
Localization analysis of the DWV-produced EGFP signal in infected honey bee brood. (**a**) Schematic representation of DWV_E_ RG construct design; only part of the viral genome, where the insert was placed, is shown; amino acid sequences are presented using single-letter code. (**b**) Combined image of fluorescent and white-field photo showing not-injected (“Mock”) and DWV_E_ injected pupae. (**c**) Confocal microscopy analysis of EGFP signal localisation in DWV_E_ infected honey bee pupae 22 h post-inoculation: dorsal side of the head with removed cuticle (“he”), thorax muscles (“mu”), wing rudiment (“wi”), crop (“cr”), Malpighian tubules (“mp”), ventriculus (“ve”), small gut (“sg”), and rectum (“re”); composite of fluorescent signal z-stack and inverted white-field image are shown for convenience of interpretation; scale bars correspond to 500 μm on all panels except “mp”, where 200 μm is shown. (**d**) Confocal microscopy analysis of living larva (first three panels) and cryosection of infected larva (right panel) sampled 6 days after feeding with 5 × 10^7^ GE of DWV_E_ inoculate: head (“he”), wing rudiments in thoracic segments (“th”), spiracle openings (“sp”), sagittal section of larva showing rudimental ovary (“ov”) and part of the midgut wall (“gt”); composite of fluorescent signal z-stack and white-field image (inverted on the cryosection panel) are shown; scale bars correspond to 500 μm.

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
