# Peer review of "Green Bees: Reverse Genetic Analysis of Deformed Wing Virus Transmission, Replication, and Tropism"

_viruses, 2020, doi:10.3390/v12050532_

Round 1

Reviewer 1 Report

This manuscript describes the development of a reverse genetic system to study the in vivo replication of Deformed wing virus (DWV) by creating constructs that express DWV genomes composed of regions from either DWV-B (formerly known as Varroa destructor virus) and DWV-A. In addition, the authors produce a recombinant DWV genome that expresses EGFP and allows for tracking of DWV infection in living larvae and pupae. This complements the work of Ryabov et al. (2020) that also produced a EGFP tagged DWV genome that was studied in infected cells. The work described in this manuscript represents an important contribution to the field and advances our ability to study the molecular biology of DWV replication, transmission, and pathogenesis.

Overall this is a clear and strong paper that uses well designed experiments with the appropriate controls. I have two major/moderate concerns about the interpretation of the data. The remaining comments are minor and are intended to improve this manuscript for publication.

Major Concerns:

  1. The experiments conducted to study replication of DWV in mites are partially dependent on the novel feeding system described in an unpublished manuscript (Christie et al., in preparation). While the interpretation of the data in Figure 2 is not completely dependent on the details of the mite feeding system, it may be prudent to either include this data in that publication or wait until that publication is in press.

In addition, the data as shown in Figure 2 does not support the claims made in the text as strongly as the authors might like. Only 2 pools of mites infected with the RG VVD virus of the 4 tested show a strong presence of a PCR band indicative of the negative strand genome. In addition, they have very faint evidence of cleavage of this band by the restriction enzyme suggesting that the majority of the virus in the mites is wild-type in origin.  For the other two mite pools the band demonstrating the presence of negative strand genome is very faint – and becomes invisible in a printed copy of the manuscript. Only after viewing the pdf copy of the manuscript could I see the bands discussed in the Results (lines 284 – 292). In one of these pools there is a stronger band at the digested mobility suggesting the majority of the replicating virus in that pool was from the recombinant genome.

The control of treating the virus inoculum with RNAseA prior to infection was important to prevent introduction of naked viral RNA (positive or negative) into the mites during feeding. But because the majority of the negative strand virus measured in their assay likely did not originate from their inoculum (as shown by the limited restriction enzyme digestion in the VVD infected mites) the authors cannot claim that this is strong evidence of DWV replication in mites. It is quite possible that mites demonstrating a strong PCR band for negative RNA just ingested the negative RNA when they fed on pupae before being collected for this experiment. While this data is suggestive of replication, I don’t believe in its current form it is strong enough to warrant the author’s claims. The major claims of this paper do not depend on this experiment and I suggest removing this data from the manuscript until it can be repeated with more convincing results.  At the very least, remove the word “strong” (line 292) from the description of this data.

2. The second major concern is the interpretation of the replication data in the Discussion. The authors seem reluctant to suggest that there are differences in the replication abilities between the different recombinant viral genomes as suggested in Figures 1, S3 and S4. While admittedly this may be preliminary data and the authors don’t want to put too much stock in the differences in replication, it is intriguing that the VDD variant, which is the only one that encodes the capsid proteins from DWV-A, appears to replicate more slowly than the two variants that contains the structural region from DWV-B. This might suggest that differences in capsid structures could influence binding and entry differences between these two viruses. The authors should address this possibility in the Discussion.

Minor Suggestions:

  1. Some of the experiments described in the Materials and Methods could be more clearly explained to help with the interpretation of the data presented.
    • The sequences of the three recombinant genomes should be deposited in Genbank so that it is clear exactly which regions of each master variant of DWV are included in each viral genome.
    • Line 109 – the EGFP encoding virus is referred to as DWVE later in the manuscript. Add that abbreviation here for clarity.
    • Lines 110-112 - While the sequence of the protease cleavage site is given in Figure 4, it could also be described in the Methods or refer to Figure 4.
    • More details on the plasmid backbone used to construct the RG constructs and which restriction enzyme was used to linearize the plasmid.
    • Line 152 - For the control genome used in the Mock infections – where exactly was the genome truncated? And was this used for all of the Mock infections described in the paper? This is not as clear as it could be.
    • Line 165 – was the cDNA reaction carried out with oligo dT or random hexamers.
    • Line 168 – what was the volume of the cDNA reaction? Was 2μL 10% of the reaction or a different fraction?
    • Line 185 - The supplementary table containing the sequence of the primers should be mentioned at the start of the Methods section for clarity.
    • Line 196 – mention that the microscopy was used to detect EGFP in samples infected with the DWVE
  2. Figure S1 is important for understanding the structures of the viral genomes tested in this paper. Instead of designating the genomes based on the original VVD recombinant – it might be clearer to designate the genomes by which regions of DWV-A and DWV-B they contain (dark for B, light for A).
  3. There are some places where additional references should be included or cited:
    1. Line 51 – should cite reference 15 (Lanzi et al, 2006) for viral classification.
    2. Line 295 should include reference 31 (Lamp et al.), which demonstrated that a recombinant viral genome deformed wings in infected bees.
    3. Line 307 – others have shown that normal looking bees can harbor high titers of DWV. These references should be cited here.
    4. Lines 370 and 485 – cite Shah et al. (Virology J, 2009) showed that DWV can replicate in the brains of infected honey bees using in situ hybridization.
  1. Minor typographical/writing errors:
    1. Line 340 – appears that the end of this sentence got deleted. The phrase “the system” doesn’t include what system is being referred to.
    2. Lines 350/353 – “mt” is used in Figure 4 legend to refer to Malpighian tubules but “mp” is used in Figure 4c.
    3. Line 360 – remove the phrase “within observable”.
    4. Line 407 – This paper did not test a full-length DWV-A genome as the 5’ UTR in all three variants is from DWV-B. Clarify for accuracy.
    5. Line 494 – sentence describing “payload” should be more clearly explained.

Author Response

Major Concerns:

  1. The experiments conducted to study replication of DWV in mites are partially dependent on the novel feeding system described in an unpublished manuscript (Christie et al., in preparation). While the interpretation of the data in Figure 2 is not completely dependent on the details of the mite feeding system, it may be prudent to either include this data in that publication or wait until that publication is in press.

The reviewer is correct that we have only provided limited information on the in vitro Varroa feeding system. This is because it forms part of a much larger manuscript to be published by Christie et al. The reviewer is also absolutely correct in the view that the interpretation of this part of the study is not completely dependent on the feeding system we used. The key feature of this part of the study was a) the use of a genetically tagged virus, and b) the demonstration that the input genetically tagged virus was free of negative strand replication intermediates. In fact, researchers who wish to recapitulate our results would be able to use a genetically tagged virus (which we do provide full details for) and prepare negative-strand RNA-free input virus stocks (which we also provide methods for) in combination with one of the already-published methods of maintaining Varroa mites in vitro. We therefore do not think it is necessary to provide exhaustive details of the maintenance system. Nevertheless, we have provided additional information to clarify our experimental design (lines 175-180).

In addition, the data as shown in Figure 2 does not support the claims made in the text as strongly as the authors might like. Only 2 pools of mites infected with the RG VVD virus of the 4 tested show a strong presence of a PCR band indicative of the negative strand genome. In addition, they have very faint evidence of cleavage of this band by the restriction enzyme suggesting that the majority of the virus in the mites is wild-type in origin. For the other two mite pools the band demonstrating the presence of negative strand genome is very faint – and becomes invisible in a printed copy of the manuscript. Only after viewing the pdf copy of the manuscript could I see the bands discussed in the Results (lines 284 – 292). In one of these pools there is a stronger band at the digested mobility suggesting the majority of the replicating virus in that pool was from the recombinant genome.

The control of treating the virus inoculum with RNAseA prior to infection was important to prevent introduction of naked viral RNA (positive or negative) into the mites during feeding. But because the majority of the negative strand virus measured in their assay likely did not originate from their inoculum (as shown by the limited restriction enzyme digestion in the VVD infected mites) the authors cannot claim that this is strong evidence of DWV replication in mites. It is quite possible that mites demonstrating a strong PCR band for negative RNA just ingested the negative RNA when they fed on pupae before being collected for this experiment. While this data is suggestive of replication, I don’t believe in its current form it is strong enough to warrant the author’s claims. The major claims of this paper do not depend on this experiment and I suggest removing this data from the manuscript until it can be repeated with more convincing results. At the very least, remove the word “strong” (line 292) from the description of this data.

Mites used in the Varroa replication study were harvested from hives with a high mite and DWV load. These mites have previously fed on pupae with a high viral load (according to our screening at the start of the field season honey bee samples taken from these colonies have shown up to 109 genome equivalents of DWV per μg of RNA). To test for DWV replication these mites were incubated on feed packets spiked with genetically tagged recombinant virus. It is therefore entirely expected that the endogenous DWV population will be amplified during subsequent analysis. However, the specific identification of negative strands containing the genetic tag (that are absent in the input virus inocula and absent from the endogenous DWV) does, in our view, provide strong evidence that DWV replicates in Varroa (as discussed in lines 488-493 of the manuscript). We acknowledge that the signal (the amount of the PCR product) is not strong. Further studies will be required to determine whether this is 1) because virus replication in mites is very slow and results in low levels of negative DWV RNA strand, 2) because mites only take limited amounts of food from feed packets or 3) because the endogenous virus already present (from the previous pupae the mite fed upon) provides strong competition to any new virus acquired by the mite (or a combination of these or other factors). These are all interesting questions, outside the scope of this study, but – because of the experimental methods described in this paper – now largely possible.

  1. The second major concern is the interpretation of the replication data in the Discussion. The authors seem reluctant to suggest that there are differences in the replication abilities between the different recombinant viral genomes as suggested in Figures 1, S3 and S4. While admittedly this may be preliminary data and the authors don’t want to put too much stock in the differences in replication, it is intriguing that the VDD variant, which is the only one that encodes the capsid proteins from DWV-A, appears to replicate more slowly than the two variants that contains the structural region from DWV-B. This might suggest that differences in capsid structures could influence binding and entry differences between these two viruses. The authors should address this possibility in the Discussion.

Slightly lower potency of VDD accumulation in inoculated honey bee brood is indeed an interesting result. This effect is particularly well seen in larvae feeding, where higher concentration of VDD inoculum was required to obtain highly infected samples. In pupae, however, the effect was only observed at lowest dilutions of the virus, when only one virus per pupa was injected. The potential differences between capsid proteins of DWV A and B could explain lower infectivity of VDD inoculate, but other interpretations, including unique internal RNA structures, are possible. As advised, we have added the corresponding discussion in lines 476-480 of the manuscript.

Minor Suggestions:

Some of the experiments described in the Materials and Methods could be more clearly explained to help with the interpretation of the data presented.

The sequences of the three recombinant genomes should be deposited in Genbank so that it is clear exactly which regions of each master variant of DWV are included in each viral genome.

Data on sequences of VVV, VVD and VDD clones was included in another manuscript submitted earlier for publication by our group and cited here as a preprint: Gusachenko, O.N.; Woodford, L.; Balbirnie-Cumming, K.; Ryabov, E.V.; Evans, D.J. Deformed Wing Virus spillover from honey bees to bumble bees: a reverse genetic study. bioRxiv 2020, 2019.12.18.880559.

We therefore initially provided a more brief description in our submission to the Viruses journal. We acknowledge that a more detailed explanation on how the DWV clones were made would be helpful. Therefore, we have added the related information on all reverse genetics construct in the revised version of the paper (lines 104-132, new Figure 1) and supplied the full cDNA sequences of all reverse genetics clones in Text S1 of supplementary material. Sequences were also submitted to the online GenBank depository (GenBank accession numbers: DWV-VDD - MT415949, DWV-VVD - MT415950, DWV-VVD_truncated - MT415951, DWV-VVV - MT415952, DWV-VDD-eGFP - MT415948, DWV-VVD-eGFP - MT415953).

Line 109 – the EGFP encoding virus is referred to as DWVE later in the manuscript. Add that abbreviation here for clarity.

Abbreviation added (line 133).

Lines 110-112 - While the sequence of the protease cleavage site is given in Figure 4, it could also be described in the Methods or refer to Figure 4.

Details on the cleavage site were added to the Materials and Methods section (lines 135-136).

More details on the plasmid backbone used to construct the RG constructs and which restriction enzyme was used to linearize the plasmid.

Details on restriction sites added (lines 118-120, 144-146 and supplementary Text S1).

Line 152 - For the control genome used in the Mock infections – where exactly was the genome truncated? And was this used for all of the Mock infections described in the paper? This is not as clear as it could be.

Sentences extended and added to clarify the experimental design (line 192, 196-197). Truncated RNA transcripts were used as one of the controls (along with PBS-injected (mock) and non-injected pupae) in RNA injections experiments only. Since very high amounts of RNA were inoculated this was required to account for possible amplification of PCR products from the injected material. Our results confirmed that pupae injected with non-functional (truncated) RNA did not produce high levels of DWV signal in qPCR and RT-PCR assays. In virus inoculation experiments (honey bee pupae injections and larvae feeding, Varroa mites feeding) mock groups were subjected to PBS solution free of the virus and RNA.

Line 165 – was the cDNA reaction carried out with oligo dT or random hexamers. Line 168 – what was the volume of the cDNA reaction? Was 2μL 10% of the reaction or a different fraction?

qScript Mastermix used in our assays contained both oligo dT and hexamer primers. 20 μl reaction volume (indeed, resulting in 2 μl constituting 10% of it) were used. Corresponding sentence in the manuscript was extended to clarify the procedure (lines 206-207).

Line 185 - The supplementary table containing the sequence of the primers should be mentioned at the start of the Methods section for clarity.

Sentence moved according to the recommendation (line 208).

Line 196 – mention that the microscopy was used to detect EGFP in samples infected with the DWVE

DWVE mention was added as advised (line 235).

Figure S1 is important for understanding the structures of the viral genomes tested in this paper. Instead of designating the genomes based on the original VVD recombinant – it might be clearer to designate the genomes by which regions of DWV-A and DWV-B they contain (dark for B, light for A).

An updated version of Figure S1 (now included in the main text as Figure 1) was submitted as recommended.

There are some places where additional references should be included or cited:

Line 51 – should cite reference 15 (Lanzi et al, 2006) for viral classification.

Citation added (line 51).

Line 295 should include reference 31 (Lamp et al.), which demonstrated that a recombinant viral genome deformed wings in infected bees.

Citation added (line 340).

Line 307 – others have shown that normal looking bees can harbor high titers of DWV. These references should be cited here.

Citations added (line 353).

Lines 370 and 485 – cite Shah et al. (Virology J, 2009) showed that DWV can replicate in the brains of infected honey bees using in situ hybridization.

Citation added (line 541).

Minor typographical/writing errors:

Line 340 – appears that the end of this sentence got deleted. The phrase “the system” doesn’t include what system is being referred to.

Corrected.

Lines 350/353 – “mt” is used in Figure 4 legend to refer to Malpighian tubules but “mp” is used in Figure 4c.

Corrected.

Line 360 – remove the phrase “within observable”.

Corrected.

Line 407 – This paper did not test a full-length DWV-A genome as the 5’ UTR in all three variants is from DWV-B. Clarify for accuracy.

Clarified as advised (line 456).

Line 494 – sentence describing “payload” should be more clearly explained.

Sentence extended as advised (line 551).

Reviewer 2 Report

DWV is one of the most damaging pest the honey bee has to face. This study is a great step towards a better understanding of DWV pathogenesis. Through reverse genetics tools, authors study host-vector-pathogen interactions (i.e. honey bee, varroa and DWV). They demonstrate DWV replication within the mite, and investigate the kinetics of virus replication at different pupal stages of infection. They also address the horizontal per os DWV transmission to young larvae.

Using a reporter gene (Green Fluorescent Protein) to study DWV tissue-tropism, they show amazing pictures of fluorescence in a wide range of tissues and organs, including nascent wing buds in DWV-fed larvae.

I highly recommend this study to be published, however I have one major and many minor points that I would like to see addressed prior to publication, both in the material and methods (which does not allow in this current version to reproduce the experiments), and in the results and discussion.

The major point is that we do not have the sequences of the constructs from this study. They need to be provided as supplementary material, and their percentage of similarity with the initial corresponding sequences (whatever recombinant or parental) have to be shown and discussed. What are the inserts sequence that were obtained by custom gene synthesis? Where exactly were introduced the restriction sites? The use is to provide all sequences with a legend about the origin of the sequence and the restrictions sites.

Nomenclature of the constructs has to be simplified, because it is very difficult to read mixed DWV/VDV names in an authors’own nomenclature, not referring to the current taxonomy of DWV-A, DWV-B and DWV-C. For example, we have to wait line 211 to start understanding what the authors mean with “VVD”, and there are a lot of abbreviations for the construct names that make sense only for the authors. L. 98: what are VDD & VVV? L.99 VDV-1VVD=?

  1. Material and Methods:

In vitro gene synthesis for DWV-A capsid proteins and DWV-B non structural proteins + amplification (even by Phusion DNA polymerase) could induce mutations. Authors should give the percentage of similarity of the plasmids (L.112-113) with the initial isolate sequences, and describe which and where the restriction sites were introduced (L. 106-108).

Sequence of the truncated VVD transcript used as negative control has also to be provided (L. 152).

2.3, 2.4: Were the injected pupae and the honey bees DWV-free before the injection? And more generally, virus-free? Possible interactions with other co-infecting viruses have to be discussed.

An injection control (PBS injection) should be added to the “mock” without inoculation to make sure that the injection does not induce per se DWV increasing levels as it has been shown that piercing honey bee cuticle can challenge their immunity (Evans et al., 2006; Siede et al., 2012; Alaux et al., 2014).

Evans, J. D., et al. (2006). "Immune pathways and defence mechanisms in honey bees Apis mellifera." Insect Molecular Biology 15(5): 645-656.

Siede, R., et al. (2012). "Comparison of transcriptional changes of immune genes to experimental challenge in the honey bee (Apis mellifera)." Journal of Apicultural Research 51(4): 320-328.

Alaux, C., et al. (2014). "Parasitic and immune modulation of flight activity in honey bees tracked with optical counters." The Journal of Experimental Biology 217(19): 3416-3424.

What about varroa mites infection with common honey bee viruses (DWV, but also ABPV, SBV, …) if collected from infested colonies (§2.5)?

L.141: feeding diet (to be published elsewhere): who, when and where? This experimental point is crucial for transmission experiments and to study virus replication into the mites and should be published before this manuscript.

Minor points:

L.112: nucleotides 1785-1786 of DWV cDNA GenBank HM067438.1?

L.144: “mites were collected and RNA was extracted” Immediately? After freezing? Using which protocol: Tri-Reagent(L.144) or GeneJet RNA Purification Kit (L.163)?

L.182: VVD9 cDNA qPCR standard sequence has to be provided and also DWVE cDNA qPCR standard sequence

2. Results

A schematic representation of all the constructs in the main text would help a lot (L. 208-211:). Figure S1 is not clear: what does mean the diferent colors? The use of black and white boxes for DWV-B and DWV-A may help to undestrand the constructs by themselves, without referring to the text. VVD-trunc and DWVE should be on the figure. The positions of the restriction sites that were introduced have to be marked by different color codes.

Injection to pupae: there is the need of an injection control. Mock with uninoculated pupae does not reflect the impact of the injection alone, that requires a buffer-injection control (see ref. above)

L.299: results are not shown, how many white-eyed pupae were inoculated? “75-80% of bees developing overtly deformed wings” seems not to be true referring to Figure 3a, where half of the bees show deformed wings, including the controls. The same was observed Fig. S4 with 45% only of the “mock” bees showing normal wings. Why control bees with such low DWV levels show wing deformities may be discussed: perhaps a physiological consequence of artificial rearing versus natural rearing in the colony, and nothing to do with DWV?

Minor points:

L.214-15: restriction tags used for unambiguous identification have to be listed.

Figure S2: only 4 pupae/construct? Were the experiments repeated?

L.227-228, 252-262 and Figure 1: Statistic tests need to be added

Figure 1: “injection or feeding” instead of “inoculation” should remove any ambiguity with varroa inoculation

3. Discussion

L.415-18: authors argue that virus populations injected by the foundress mite may be predominant in all progeny mites. However how they explain why some mites (Figure 2) exhibited the wild type of DWV (not digested) and not the VVD fed DWV?

Oral virus transmission may be compared to the bibliography: are 106-107 GE of virus far from the doses already used by other authors? (Möckel et al., 2011; Iqbal and Muller, 2007; Khongphinitbunjong et al., 2015 ; Thaduri et al., 2019 ;  Ryabov et al., 2016)

Möckel, N., et al. (2011). "Horizontal transmission of deformed wing virus: pathological consequences in adult bees (Apis mellifera) depend on the transmission route." Journal of General Virology 92(2): 370-377.

Iqbal, J. and U. Mueller (2007). "Virus infection causes specific learning deficits in honeybee foragers." Proc Biol Sci 274(1617): 1517-1521.

Khongphinitbunjong, K., et al. (2016). "Responses of Varroa-resistant honey bees (Apis mellifera L.) to Deformed wing virus." Journal of Asia-Pacific Entomology 19(4): 921-927.

Thaduri, S., et al. (2019). "Disentangling host-parasite-pathogen interactions in a varroa-resistant honeybee population reveals virus tolerance as an independent, naturally adapted survival mechanism." Scientific Reports 9(1): 6221.

Ryabov, E. V., et al. (2016). "The Iflaviruses Sacbrood virus and Deformed wing virus evoke different transcriptional responses in the honeybee which may facilitate their horizontal or vertical transmission." Peerj 4.      

The input of putative transmission by the queen should be discussed L.423-427 and 456-459. Virus-feeding is not the unique virus source if the queen has already transmitted DWV through the egg (see Amiri E, Kryger P, Meixner MD, Strand MK, Tarpy DR, et al. (2018) Quantitative patterns of vertical transmission of deformed wing virus in honey bees. PLOS ONE 13(3): e0195283. https://doi.org/10.1371/journal.pone.0195283).

Last, a single pure molecular clone, in this study with some part of the genome originating from gene synthetis, should be compared to the natural quasi-species virus populations: are the constructs representing the genetic diversity of the original virus population? What functional differences can be expected from more complex natural quasi-species ?(see Yañez, O., et al. (2020). "The honeybee (Apis mellifera) developmental state shapes the genetic composition of the deformed wing virus-A quasispecies during serial transmission." Scientific Reports 10(1): 5956.)

Minor revisions (introduction):

  1. 40 : rather than citing [2,3], cite the recent review about virus distribution (Beaurepaire, A., et al. (2020). "Diversity and Global Distribution of Viruses of the Western Honey Bee, Apis mellifera." Insects 11(4): 239.)

L.83-84 : RG is a standard approach but needs cell culture or at least healthy individuals to be used.

incomplete reference 9. Ryabov, E.V.; Childers, A.K.; Lopez, D.; Grubbs, K.; Posada-Florez, F.; Weaver, D.; Girten, W.;vanEngelsdorp, D.; Chen, Y.; Evans, J.D. Dynamic evolution in the key honey bee pathogen Deformed wing virus :

L.57: 3 groups, not 2. DWV-C has been described in the UK (Kevill, J. L., et al. (2017). "ABC Assay: Method Development and Application to Quantify the Role of Three DWV Master Variants in Overwinter Colony Losses of European Honey Bees." Viruses 9(11); Mordecai, G. J., et al. (2015). "Diversity in a honey bee pathogen: first report of a third master variant of the Deformed Wing Virus quasispecies." ISME J.)

Author Response

The major point is that we do not have the sequences of the constructs from this study. They need to be provided as supplementary material, and their percentage of similarity with the initial corresponding sequences (whatever recombinant or parental) have to be shown and discussed. What are the inserts sequence that were obtained by custom gene synthesis? Where exactly were introduced the restriction sites? The use is to provide all sequences with a legend about the origin of the sequence and the restrictions sites.

Data on composition of VVV, VVD and VDD clones of DWV was previously included in another manuscript submitted for publication by our group and cited here as a preprint: Gusachenko, O.N.; Woodford, L.; Balbirnie-Cumming, K.; Ryabov, E.V.; Evans, D.J. Deformed Wing Virus spillover from honey bees to bumble bees: a reverse genetic study. bioRxiv 2020, 2019.12.18.880559.

We acknowledge that a more detailed explanation on the reverse genetics DWV clones composition would be helpful. We therefore supplied the manuscript with the supplementary text S1 containing full sequences of our clones cDNA and included in the main manuscript body text a description of the location of the unique genetic tags used to differentiate the virus genomes. As wisely suggested by the reviewer we have additionally colour-coded clones sequences in the supplementary text to make it clear where individual parts of the recombinant virus genomes originate from (also illustrated on the updated Figure 1).

Nomenclature of the constructs has to be simplified, because it is very difficult to read mixed DWV/VDV names in an authors’own nomenclature, not referring to the current taxonomy of DWV-A, DWV-B and DWV-C. For example, we have to wait line 211 to start understanding what the authors mean with “VVD”, and there are a lot of abbreviations for the construct names that make sense only for the authors. L. 98: what are VDD & VVV? L.99 VDV-1VVD=?

We chose to preserve the three letter abbreviated nomenclature (as VDD, VVV and VVD) used in our manuscript as these names reflect the origin of the clones sequences and the names of the strain types as recorded in GenBank. This nomenclature was historically established in our laboratory and already included in several published manuscripts. Letters V and D refer to the sequence homology with Varroa destructor virus 1 (DWV B) and DWV A respectively. The cDNA sequence of the recombinant DWV clone used in our study and named VVD throughout the text was based on the full clone of DWV variant VDV-1-DWV-No-9 (GenBank HM067438.1). This virus was erroneously labeled by us as VDV-1VVD in the initially submitted manuscript. Thanks to the reviewer's comment we thereof corrected the name in the revised submission (line 101 and further through the text). The new Figure 1 submitted with the revised manuscript shows both the strains labelled DWV A and VDV-1, colour coded, with the lines below showing our clonal viruses and their naming. Explanation of our virus clones nomenclature was also included in the text of the manuscript (lines 114-116). We hope that this update will allow clearer reading of our work.

Material and Methods:

In vitro gene synthesis for DWV-A capsid proteins and DWV-B non structural proteins + amplification (even by Phusion DNA polymerase) could induce mutations. Authors should give the percentage of similarity of the plasmids (L.112-113) with the initial isolate sequences, and describe which and where the restriction sites were introduced (L. 106-108). Sequence of the truncated VVD transcript used as negative control has also to be provided (L. 152).

All virus cDNAs were sequenced in their entirety before using them in this study (as stated in lines 137-138). Recombinant VVD clone is homologous with the source virus genomic composition (VDV-1-DWV-No-9) with the only modified region corresponding to the new restriction site (2 nucleotides changed). VDD and VVV clones were produced by substitution of VVD sequence parts which allowed to obtain 99,65% and 99,79% identity on the encoded proteins level with the reference field DWV A and B. Following the reviewer's recommendations this information and detailed description of the new restriction sites incorporated into each construct were added to the manuscript text (lines 104-110 and 256-258, the legend of the new Figure 1). The sequence of the truncated VVD product was included in supplementary Text S1.

2.3, 2.4: Were the injected pupae and the honey bees DWV-free before the injection? And more generally, virus-free? Possible interactions with other co-infecting viruses have to be discussed.

The bees and pupae were not DWV free. With the possible exception of pupae from Australia, there are no DWV-free bees as the virus is endemic in the western honey bee population. However, the levels of DWV were quantified in advance (see Figure S1) and are routinely measured in our research apiary. Other viruses may have been present, but we have no phenotypic evidence for this. Whilst we could speculate on potential interactions it is clear from our replication studies that inoculated DWV replicates extremely rapidly and reaches maximum levels (in line with those reported in other studies) within 48 hours. This suggests that, if such interactions are occurring, they are of little or no consequence to the replication of DWV. The experiments presented here are representative of a subset of those conducted using different parental colonies for the pupae/larvae in different seasons. If the results were due to interactions with other viruses these would also have to be present in other hives/seasons/experiments as well. We believe that inclusion of a comment on potential interactions, where we have no evidence for the presence of other viruses, would raise unnecessary questions for the reader and so respectfully have not modified the manuscript. However, we have included (lines 163-166) a point about the pre-screening of our colonies for experimental harvesting of honey bee samples.

An injection control (PBS injection) should be added to the “mock” without inoculation to make sure that the injection does not induce per se DWV increasing levels as it has been shown that piercing honey bee cuticle can challenge their immunity (Evans et al., 2006; Siede et al., 2012; Alaux et al., 2014).

Evans, J. D., et al. (2006). "Immune pathways and defence mechanisms in honey bees Apis mellifera." Insect Molecular Biology 15(5): 645-656.

Siede, R., et al. (2012). "Comparison of transcriptional changes of immune genes to experimental challenge in the honey bee (Apis mellifera)." Journal of Apicultural Research 51(4): 320-328.

Alaux, C., et al. (2014). "Parasitic and immune modulation of flight activity in honey bees tracked with optical counters." The Journal of Experimental Biology 217(19): 3416-3424.

We acknowledge that previous studies have demonstrated that experimental puncturing of honey bees may represent a challenge to the system. Since puncturing inevitably occurs when Varroa transmits DWV in the field, we intentionally used PBS-injected control samples to account for potential increase in DWV accumulation due to the injection procedure itself. However, possibly due to the overall low level of endogenous DWV in our experimental colonies, we did not observe DWV accumulation after PBS injections. We additionally tested whether preliminary piercing (4 or 20 h prior to injection with the virus) affects subsequently introduced virus accumulation rates and found no evidence of the positive or negative effect of this procedure. Following the reviewer's recommendation we added a new supplementary figure (Figure S1) to the revised version of the manuscript showing analysis of endogenous (‘field’) DWV level in PBS-injected and non-injected pupae from one of our colonies. Pupae were tested 24 h and 8 days post-treatment (mock-injection with PBS) in order to verify whether any short- or long-term effect of the procedure could be seen. Apart from one bee (out of 13 samples), which demonstrated high DWV titer (and developed with characteristic DWV deformities), none of the samples show elevated levels of the virus. It, therefore, appears more likely that the only bee which developed with high DWV was infected in the colony prior to the experiment rather than an induced infection due to the mock injection procedure.

What about varroa mites infection with common honey bee viruses (DWV, but also ABPV, SBV, …) if collected from infested colonies (§2.5)?

We did not perform the screening of Varroa mites used for the feeding packet experiment for any viruses other than DWV. Therefore, we cannot completely exclude the possibility of competition between the fed clonal DWV and other viruses within the mite organism. However, we have no phenotypic evidence for overt levels of sacbrood, acute (or other) paralysis viruses in colonies used for mite samplings. If any viruses other than DWV were present in these colonies they were not associated with symptomatic infection and hence most probably persisted at very low levels.

L.141: feeding diet (to be published elsewhere): who, when and where? This experimental point is crucial for transmission experiments and to study virus replication into the mites and should be published before this manuscript.

We do not feel that this is essential for the replication in Varroa study as mite feeding packet systems have already been published. What is unique in our study is the combination of a) use of a genetically-tagged recombinant input virus b) pre-treatment of the virus inocula with RNAse A to remove contaminating negative strand replication intermediates c) following detection of the tagged negative strand product. Using exactly this approach, combined with an already-published varroa feed-packet system (such as Cabrera et al (2017), Egekwu et al (2018), Jack et al (2020), preprint by Posada-Florez et al (2020)) would enable the essential results of the study to be recapitulated. The honey bee free feed packet system is being published separately by our collaborators as part of a much larger study on the replication of Varroa. However, following the reviewers recommendation we now include additional information on the mite feeding in the Materials and Methods section (lines 174-180).

Minor points:

L.112: nucleotides 1785-1786 of DWV cDNA GenBank HM067438.1?

Clarified in lines 105-106 of the revised manuscript.

L.144: “mites were collected and RNA was extracted” Immediately? After freezing? Using which protocol: Tri-Reagent(L.144) or GeneJet RNA Purification Kit (L.163)?

Clarified (lines 183-184).

L.182: VVD9 cDNA qPCR standard sequence has to be provided and also DWVE cDNA qPCR standard sequence

We agree that the provided description of qPCR standard preparation was not quite clear. For qPCR standards, we used serial dilutions of cDNA obtained with 1 μg of RNA transcribed from VVD and DWVE linearized templates (same RNA which was used for injections into pupae to obtain the virus inoculates). The same primers sets were then used in qPCR both for standard curve samples and experimental samples. An adequate description of this procedure was added in lines 224-226 of the revised text.

  1. Results

A schematic representation of all the constructs in the main text would help a lot (L. 208-211:). Figure S1 is not clear: what does mean the diferent colors? The use of black and white boxes for DWV-B and DWV-A may help to undestrand the constructs by themselves, without referring to the text. VVD-trunc and DWVE should be on the figure. The positions of the restriction sites that were introduced have to be marked by different color codes.

As recommended by the reviewer, we have moved the Figure S1 to the main text of the manuscript (now Figure 1). We did our best to provide all required details regarding the clones composition and to address all of the reviewer's comments above. This data can now be found on Figure 1 and in the caption below.

Injection to pupae: there is the need of an injection control. Mock with uninoculated pupae does not reflect the impact of the injection alone, that requires a buffer-injection control (see ref. above)

We agree that based on the previously published data and common sense it is necessary to take into account the impact of the injections itself. This has been also addressed by several previously published works, for example, in manuscript by Tehel et al from 2019. Hence, we found it excessive to provide repeating data within already voluminous publication. We can confirm that both injection and laboratory handling had an impact on the resulting morbidity rates in our pupae development experiments. We found that non-injected pupae developed under the same conditions have shown lower (although still high) levels of developmental abnormalities with 60 to 75% of bees eclosing normally compared to 60% in the PBS injected group. However, we also consider this control to be less relevant to the aim of our experiments on pupae inoculations - which was to find whether morbidity rates differed between the clonal DWV variants. In this case it appeared more appropriate to include injected pupae as a control group in order to demonstrate that the observed morbidity rate was distinguishable from that caused by injection and handling itself. As mentioned above, influence of the injection on DWV accumulation was also tested (data added to the revised manuscript as Figure S1).

L.299: results are not shown, how many white-eyed pupae were inoculated? “75-80% of bees developing overtly deformed wings” seems not to be true referring to Figure 3a, where half of the bees show deformed wings, including the controls. The same was observed Fig. S4 with 45% only of the “mock” bees showing normal wings. Why control bees with such low DWV levels show wing deformities may be discussed: perhaps a physiological consequence of artificial rearing versus natural rearing in the colony, and nothing to do with DWV?

Indeed, the number of samples in each group is not specified in the text of the manuscript. However, this information can be found in the legends of the discussed figures (line 361 and description below the Figure S4). Figure 3a (Figure 4a in the revised manuscript) is showing qPCR analysis data of 4-6 samples with clearly deformed and normal phenotypes selected from each of the treatment groups. Meanwhile, Figure 3b (4b in the revised manuscript) shows the quantitative distribution of developed pupae according to displayed symptoms. We accept that the initial citing of the figure panels in the text was a bit vague and following the reviewers comment we have revised the corresponding sentence to add clarity (lines 344-346). We also agree with the reviewer’s notion that the observed levels of developmental deformities are sufficiently high to conclude that development of these bees was affected by the experimental handling. We added discussion of the subject in lines 376-380.

Minor points:

L.214-15: restriction tags used for unambiguous identification have to be listed.

All restriction tags introduced in the virus cDNA are now listed in lines 118-120.

Figure S2: only 4 pupae/construct? Were the experiments repeated?

Figure S2 shows qPCR analysis results for 4 out of 8 samples (from one experiment) injected with in vitro synthesized viral RNA. In this experiment all samples injected with full-length RNA (8 samples for each of the three clones - VVV, VDD or VVD) produced a high intensity band showing the presence of DWV RNA when screened by end-point RT-PCR. The experiment was repeated multiple times, since all DWV stocks used in our laboratory throughout a three-years long project were obtained via this procedure. We used RNA-injected pupae rather than passaging of the virus in order to reduce potential accumulation of sequence mutations within new virus generations. In our hands recovery of DWV from RNA transcript injected honey bee pupae has proven to be efficient and reproducible.

L.227-228, 252-262 and Figure 1: Statistic tests need to be added

Added (lines 271 and 301).

Figure 1: “injection or feeding” instead of “inoculation” should remove any ambiguity with varroa inoculation

Revised.

  1. Discussion

L.415-18: authors argue that virus populations injected by the foundress mite may be predominant in all progeny mites. However how they explain why some mites (Figure 2) exhibited the wild type of DWV (not digested) and not the VVD fed DWV?

The sentence under question refers to the ability of the DWV variants directly introduced into the honey bee pupa by the mite to replicate very quickly. In contrast, the results of the experiment on Varroa feeding on Figure 2 (Figure 3 in the revised version of the manuscript) originate from a different situation where the virus replicates in mites only with no honey bee host involved. It is clear that the ingested virus replicates much better in bees than mites which explains the observed results.

Oral virus transmission may be compared to the bibliography: are 106-107 GE of virus far from the doses already used by other authors? (Möckel et al., 2011; Iqbal and Muller, 2007; Khongphinitbunjong et al., 2015 ; Thaduri et al., 2019 ; Ryabov et al., 2016)

We thank the reviewer for the proposed reference suggestions. The works by Möckel et al. (2011) and Iqbal and Muller (2007) describe oral infection of adult bees, which may have different susceptibility to the virus compared to larval stage. While the publication by Khongphinitbunjong et al. (2015) provides more insight into DWV infectivity in larvae this work describes feeding with serial dilutions of wild-type DWV and to the best of our knowledge does not provide quantitative data on the virus titers. According to Thaduri et al. (2019) and Ryabov et al. (2016) the authors used high concentrations of the virus for oral larvae inoculations (108-1010 compared to 104-108 range tested in our study) in order to obtain guaranteed infection. Additionally, different experimental design of these works (in particular, mixed infection with sacbrood virus) and use of mixed-population DWV inoculum does not allow to perform any direct comparison with our results.

The input of putative transmission by the queen should be discussed L.423-427 and 456-459. Virus-feeding is not the unique virus source if the queen has already transmitted DWV through the egg (see Amiri E, Kryger P, Meixner MD, Strand MK, Tarpy DR, et al. (2018) Quantitative patterns of vertical transmission of deformed wing virus in honey bees. PLOS ONE 13(3): e0195283. https://doi.org/10.1371/journal.pone.0195283).

We agree with the reviewers comment on the importance of the impact of vertically transmitted DWV. Certainly, this route should have an influence on the colony's health, especially in the case of a highly infected queen. Our data does not provide any direct insights into this process and hence does not allow discussing this subject within the framework of the manuscript. As advised, we have added a suggested reference with a sentence in lines 512-515.

Last, a single pure molecular clone, in this study with some part of the genome originating from gene synthetis, should be compared to the natural quasi-species virus populations: are the constructs representing the genetic diversity of the original virus population? What functional differences can be expected from more complex natural quasi-species ? (see Yañez, O., et al. (2020). "The honeybee (Apis mellifera) developmental state shapes the genetic composition of the deformed wing virus-A quasispecies during serial transmission." Scientific Reports 10(1): 5956.)

The coding sequences of VDD and VVV clones used in our work bear 85.05% identity between each other that is close to the sequence homology of the field DWV A and B (84.54%). We therefore believe that our experimental system represents a sufficiently broad range of sequence variation mimicking the field populations of the virus. Whilst acknowledging the importance of viral quasispecies to the biology of positive strand RNA viruses, we feel that this question is far more deserving of a study of its own. It is a fundamentally different question to those we have addressed in this report. To expand this manuscript with comparative studies of clonal vs. divergent virus population is likely to lead to a significant increase in length of the manuscript and a potential increase in complexity to the detriment of the reader. In addition, the inclusion of these additional studies would, of necessity, delay submission of the manuscript.

Minor revisions (introduction):

40 : rather than citing [2,3], cite the recent review about virus distribution (Beaurepaire, A., et al. (2020). "Diversity and Global Distribution of Viruses of the Western Honey Bee, Apis mellifera." Insects 11(4): 239.)

Citation added.

incomplete reference 9. Ryabov, E.V.; Childers, A.K.; Lopez, D.; Grubbs, K.; Posada-Florez, F.; Weaver, D.; Girten, W.;vanEngelsdorp, D.; Chen, Y.; Evans, J.D. Dynamic evolution in the key honey bee pathogen Deformed wing virus :

Reference completed.

L.57: 3 groups, not 2. DWV-C has been described in the UK (Kevill, J. L., et al. (2017). "ABC Assay: Method Development and Application to Quantify the Role of Three DWV Master Variants in Overwinter Colony Losses of European Honey Bees." Viruses 9(11); Mordecai, G. J., et al. (2015). "Diversity in a honey bee pathogen: first report of a third master variant of the Deformed Wing Virus quasispecies." ISME J.)

Updated, citation added (line 63).